# May We Use Non-Invasive Indices of Aortic Stiffness and Endothelial Glycocalyx as Biomarkers for Idiopathic Pulmonary Artery Hypertension Follow-Up?

**DOI:** 10.3390/medicina57060558

**Published:** 2021-06-01

**Authors:** Helen Triantafyllidi, Dionysia Birmpa, Dimitrios Benas, Ignatios Ikonomidis, Antonis Schoinas, Paraskevi Trivilou, Frantzeska Frantzeskaki, Stylianos E. Orfanos

**Affiliations:** 12nd Department of Cardiology, Medical School, National and Kapodistrian University of Athens, ATTIKON University Hospital, 12462 Athens, Greece; dbirba@gmail.com (D.B.); dimitriosbenas@gmail.com (D.B.); ignoik@otenet.gr (I.I.); a_schinas@yahoo.gr (A.S.); ptrivilou@yahoo.gr (P.T.); 22nd Department of Critical Care, Medical School, National and Kapodistrian University of Athens, ATTIKON University Hospital, 12462 Athens, Greece; ffrantzeska@gmail.com (F.F.); stylianosorfanosuoa@gmail.com (S.E.O.)

**Keywords:** aortic stiffness, endothelial glycocalyx, idiopathic pulmonary arterial hypertension

## Abstract

Idiopathic pulmonary arterial hypertension (IPAH) initial evaluation and follow-up, a rare and incurable disease if left untreated, is based on a multiparametric approach (functional status of the patient, biomarkers, hemodynamic parameters and imaging evaluation of right heart impairment). Arterial stiffness (AS) and endothelial glycocalyx are indices of systemic circulation. We present the 3-years follow-up of a female IPAH patient. We propose aortic stiffness and endothelial glycocalyx indices as non-invasive markers of either improvement or deterioration of IPAH disease.

## 1. Introduction

Pulmonary arterial hypertension (PAH) represents a rare and incurable disease with increased morbidity and mortality if disease left untreated [1]. PAH is characterized by pulmonary arterial endothelial injury and cell proliferation which lead to pulmonary endothelial dysfunction, impaired vessel wall compliance, increased pulmonary vascular resistance (PVR) and reduced cardiac output [2,3]. Prognosis of patients with PAH is difficult to determine and is based on a multiparametric approach based on functional status of the patient, biomarkers, hemodynamic parameters and imaging evaluation of right heart impairment [1]. However, few studies have focused on the role of the systemic aortic stiffness (AS) and endothelial function in PAH with conflicting results regarding increased vascular compliance and endothelial dysfunction in several cohorts with Idiopathic pulmonary arterial hypertension (IPAH) or PAH-associated diseases [4,5,6,7].

Systemic AS can be accurately estimated by carotid-femoral pulse wave velocity measurement (PWV) (Complior SP, Artech Medical, Pantin, France) [8]. Increased PWV represents impaired aortic compliance. Systemic endothelial glycocalyx (EG) integrity, an important part of the endothelium, represents endothelial function. It can be estimated by the perfused boundary region measurement (PBR) of the sublingual arterial microvessels (diameter ranged 5–9 μm) using Sidestream Darkfield imaging (Microscan, Glycocheck, Microvascular Health Solutions Inc., Salt Lake City, UT, USA). Increased PBR indicates EG thickness reduction which allows blood cell penetration, loss of EG barrier properties and endothelial dysfunction [9].

## 2. Case Report

We present the case of a female patient, 51 years old, with idiopathic PAH (IPAH), followed by our PH clinic for the last 15 years. The patient had never smoked and she didn’t suffer from arterial hypertension, diabetes mellitus or hyperlipidemia. When she was first diagnosed with IPAH due to shortness of breath, the right heart catheterization (RHC) revealed a mean pulmonary artery pressure (mPAP) equal to 49 mmHg, a pulmonary vascular resistance (PVR) equals to 10 Wood Units and a cardiac index (CI) equal to 2.5 L/min/m^2^ while her six-minute walking distance (6MWD) was 570 m. The patient started treatment with bosentan while sildenafil at first and then subcutaneous treprostenil were added. The latter was progressively uptitrated recently to 60 ng/kg/min. The annual patient’s monitoring for the last 3 years (2018–2020) is being presented while she was on triple therapy (bosentan, sildenafil and subcutaneous treprostinil). We followed the recommendations by the recent guidelines exploring the role of changes of systemic circulation biomarkers as non-invasive indices of PAH severity (aortic stiffness and endothelial glycocalyx estimation).

On 2018 evaluation, the patient was in World Health Organization- Functional Class (WHO-FC) late II status. After RHC, we found that mPAP = 54 mmHg, PVR = 8.3 Wood Units and CI = 2.8 L/min/m^2^. In 6MWD she walked 561 m while in cardiopulmonary exercise test (CPET) she achieved a peak oxygen consumption (peak VO_2_) =14.7 mL/min/kg (66% predicted VO_2_). PWV was 8.30 m/s and PBR 5–9 was 1.11 μm. Simultaneously, we found that PBR 5–25 = 2.36 μm, PBR 10–19 = 2.42 μm and PBR 20–25 = 3.29 μm while red blood filling percentage (RBC%) =65%. The latter reflects the percentage of time in which a particular vascular segment was occupied by RBCs. 

On 2019 evaluation, the patient remained on WHO-FC II-late status and RHC revealed that: mPAP (55 mmHg) and CI (2.7 L/min/m^2^) remained almost stable while PVR was increased to 9.2 Wood Units. Similarly, she walked 9 m less in 6MWD (552 m) and she achieved lower results during CPET (peak VO_2_ = 12.7 mL/min/kg, 56% predicted VO_2_). Moreover, PWV was decreased to 7.8 m/sec while PBR 5–9 was increased to 1.24 μm. We also found that PBR 5–25 = 2.09 μm and PBR 20–25 = 2.19 μm were decreased, PBR 10–19 = 2.47 μm was increased while red blood filling percentage (RBC%) = 61% (decreased). Those results (mainly PVR increase) resulted to the replacement of sildenafil with riociguat which was gradually uptitrated to 2.5 mg tid. 

During the last evaluation on October 2020, we noticedthat the patient’s WHO-FC status was upgraded to II-early. RHC confirmed that her hemodynamic parameters were improved: mPAP was decreased to 48 mmHg; PVR was decreased to 6.9 Wood Units and CI was increased to 3 L/min/m^2^. Similarly, she walked 13 m more (6MWD = 565 m) and she achieved better results in CPET (peak VO_2_ = 13.7 mL/min/kg, 61% predicted VO_2_). Moreover, PWV was again increased to 8.3 m/sec while PBR 5–9 was again decreased to 1.13 μm. Additionally, PBR 5–25 = 2.22 μm and PBR 20–25 = 2.94 μm were again increased, PBR 10–19 = 2.34 μm was decreased while red blood filling percentage (RBC%) =69% (increased).

It seems that when the clinical status and the hemodynamic parameters (PVR) of this IPAH female patient were deteriorated (2nd evaluation), aortic stiffness was decreased and endothelial function worsened. On the contrary, when the patient was clinically, functionally and hemodynamically increased and endothelial function was recovered (3rd evaluation, Figure 1). This study was approved by the Institutional Review Board, ATTIKON University Hospital (protocol ID 267, submitted on 25 April 2019 and approved on 21 May 2019).

## 3. Discussion

In a recent paper, we concluded that PVR was independently and inversely predicting PWV levels (Beta = −0.46, *p* = 0.01) in a group of 31 naïve first-diagnosed PAH patients (suffering of IPAH or PAH associated with connective tissue disease, congenital heart disease or portopulmonary hypertension) [7]. Several mechanisms, besides changes in cardiac output, might be responsible for the alterations observed in systemic vascular status in IPAH disease. The high activity of circulatory mediators (pro-inflammatory cytokines) and hypoxia may lead to impaired aortic compliance and endothelial dysfunction. On the otherhand, IPAH specific treatment (i.e., phosphodiesterase type 5 inhibitors, PDE5-i, prostacyclins) might promote Nitric oxide (NO) release and endothelial NO receptors upregulation and subsequently might improve endothelial function in the systemic circulation [10,11]. 

## 4. Conclusions

Advanced IPAH therapy and improvement of hemodynamic parameters leads to in-creased cardiac output which in turn results to systemic circulation restoration. On the same time, indices of systemic circulation (aortic stiffness and endothelial glycocalyx) alter. Although this is only a case report, we think that changes regarding aortic stiffness and endothelial glycocalyx indices during the IPAH treatment need to be further studied as possible non-invasive markers of either improvement or deterioration of IPAH disease.

## Figures and Tables

**Figure 1 medicina-57-00558-f001:**
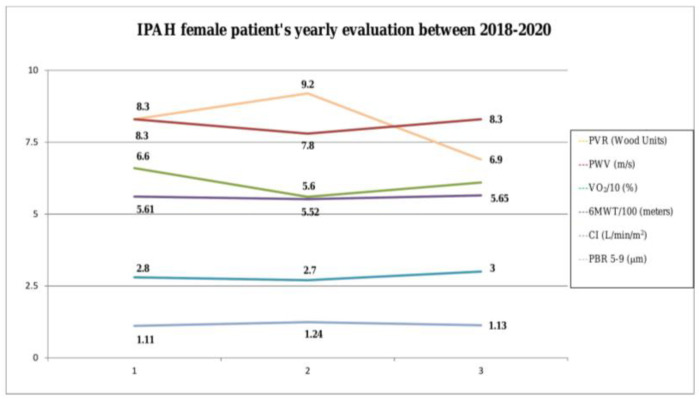
Annual changes in hemodynamic, exercise and systemic circulation indices. (Abbreviations: PVR, pulmonary vascular resistance; PWV, pulse wave velocity; VO_2_, oxygen consumption, 6MWT, six-minute walking test; CI, cardiac index; PBR, perfused boundary region).

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
