# Peer review of "May We Use Non-Invasive Indices of Aortic Stiffness and Endothelial Glycocalyx as Biomarkers for Idiopathic Pulmonary Artery Hypertension Follow-Up?"

_medicina, 2021, doi:10.3390/medicina57060558_

Round 1
Reviewer 1 Report
I have read with interest the case report by Triantafyllidi et al.. In this manuscript the authors present the case of a female patient suffering from IPAH and report a follow up of her aortic stiffness and endothelial glycocalyx measured sublingually. The language used is appropriate. However, I have some issues I would like to address:
- In a previous study by the same study group, PBR 5-9 was decreased in IPAH patients compared to healthy controls. (Triantafyllidi, Helen, et al. "Systemic endothelial glycocalyx and aortic stiffness are preserved in pulmonary arterial hypertension." Hellenic journal of cardiology: HJC= Hellenike kardiologike epitheorese 61.4 (2020): 288-290.). However, in this case report, a damaged glycocalyx is associated with disease deterioration. Can the authors further explain this discrepancy between their findings?
- To what extent could the observed changes be the direct effect of the medication (e.g. prostacyclin) on the endothelium?
- Did the patient have further comorbidities that could influence the aortic stiffness or/and glycocalyx values? What about tobacco use?
- GlycoCheck reports not only PBR 5-9, but also PBR 10-19, PBR 20-25, PBR 4-25 as well as RBC filling. Were there any changes observed in the rest of the markers?
- Regarding the fact that the authors present a case report, I would kindly suggest to downtone the conclusions of the current manuscript. The cause of the observed endothelial changes in this patient is vague and a coincidence cannot be excluded.
Author Response
Authors' answers to Reviewer #1 comments:
I have read with interest the case report by Triantafyllidi et al. In this manuscript the authors present the case of a female patient suffering from IPAH and report a follow up of her aortic stiffness and endothelial glycocalyx measured sublingually. The language used is appropriate. However, I have some issues I would like to address:
- In a previous study by the same study group, PBR 5-9 was decreased in IPAH patients compared to healthy controls. (Triantafyllidi, Helen, et al. "Systemic endothelial glycocalyx and aortic stiffness are preserved in pulmonary arterial hypertension." Hellenic journal of cardiology: HJC= Hellenike kardiologike epitheorese 61.4 (2020): 288-290.). However, in this case report, a damaged glycocalyx is associated with disease deterioration. Can the authors further explain this discrepancy between their findings?
Answer: We are very pleased that you are aware of our previous study. Of course, the previous study was a cross sectional study which referred to a totally different population, the naïve PAH patients, who were compared to normal subjects. Our findings included: a. the negative association between PVR and PWV which is also true for the present case report and b. the lower endothelium glycocalyx index (PBR 5-9) compared to normal subjects while no relationship was found between PAH severity and PBR. However, we didn’t record any changes (PWV, PBR 5-9) through PAH therapy in that previous study which actually is one of our future plans.
- To what extent could the observed changes be the direct effect of the medication (e.g. prostacyclin) on the endothelium?
Answer: Indeed, as we already mention in Discussion section, we believe that IPAH treatment based on NO increase (i.e. prostacyclin, PDE5 inhibitors) possibly acts not only in the pulmonary circulation but also in the systematic circulation and probably improves endothelial function. We will design a new study in the future regarding the role of prostacyclin regarding changes of the periphery circulation (text added).
- Did the patient have further comorbidities that could influence the aortic stiffness or/and glycocalyx values? What about tobacco use?
Answer: No, the patient had never smoked and she didn’t suffer from arterial hypertension, diabetes mellitus or hyperlipidemia (text added at the beginning of Case Report section).
- GlycoCheck reports not only PBR 5-9, but also PBR 10-19, PBR 20-25, PBR 5-25 as well as RBC filling. Were there any changes observed in the rest of the markers?
Answer: Changes similar to PBR 5-9 were observed in PBR 10-19 but not in PBR 20-25. On the other hand, PBR 5-25 is the mean of PBR 5-9, PBR 10-19 and PBR 20-25 so PBR 5-25 changes reflect the changes in the other 3 indices. In several studies in different populations under investigation, we have concluded that PBR 5-9 is the most representative index between the other 3 ones regarding endothelial glycocalyx. We present a number of previous studies of our research group:
1: Triantafyllidi H et al. I. Sex-related associations of high-density lipoprotein cholesterol with aortic stiffness and endothelial glycocalyx integrity in treated hypertensive patients. J Clin Hypertens (Greenwich). 2020Oct;22(10):1827-1834.
2: Ikonomidis I, … Triantafyllidi H et al. Impaired Arterial Elastic Properties and Endothelial Glycocalyx in Patients with Embolic Stroke of Undetermined Source. Thromb Haemost. 2019 Nov;119(11):1860-1868.
3: Triantafyllidi H, et al.HDL cholesterol levels and endothelial glycocalyx integrity in treated hypertensive patients. J Clin Hypertens (Greenwich). 2018 Nov;20(11):1615-1623.
4: Ikonomidis I, …Triantafyllidi H et al. Effects of varenicline and nicotine replacement therapy on arterial elasticity, endothelial glycocalyx and oxidative stress during a 3-month smoking cessation program. Atherosclerosis. 2017 Jul;262:123-130.
- Regarding the fact that the authors present a case report, I would kindly suggest to downtone the conclusions of the current manuscript. The cause of the observed endothelial changes in this patient is vague and a coincidence cannot be excluded
Answer: You are totally right. We believe that now our conclusions are in the right way suggesting the need for future research (text changed).
Reviewer 2 Report
The Authors of the manuscript entitled “Case Report “May we use non-invasive indices of aortic stiffness and endothelial glycocalyx as biomarkers for idiopathic pulmonary artery hypertension follow-up?” have presented a case report of a 51 years old female patient with IPAH. They propose that aortic stiffness and endothelial glycocalyx indices could be examined as non-invasive markers of either improvement or deterioration of IPAH disease. This is an interesting observation that needs to be studied further in a larger number of patients.
Lines 63- 65: You state that on 2019 evaluation, the patient remained on WHO-FC II-late status and RHC revealed the following deterioration: mPAP was increased to 55 mmHg; PVR was increased to 9.2 Wood Units while CI was decreased to 2.7 lt/ min/m2.
Lines 70-73: Here you state that during the last evaluation on October 2020, we noticed that the patient’s WHO-FC status was upgraded to II-early. RHC confirmed that her hemodynamic parameters were improved: mPAP was decreased to 48 mmHg; PVR was decreased to 6.9 Wood Units and CI was increased to 2.7 lt/ min/m2. The figure shows the CI to be 3L/min/m2. Please correct this discrepancy.
Author Response
Authors' answers Reviewer #2 comments
The Authors of the manuscript entitled “Case Report “May we use non-invasive indices of aortic stiffness and endothelial glycocalyx as biomarkers for idiopathic pulmonary artery hypertension follow-up?” have presented a case report of a 51 years old female patient with IPAH. They propose that aortic stiffness and endothelial glycocalyx indices could be examined as non-invasive markers of either improvement or deterioration of IPAH disease. This is an interesting observation that needs to be studied further in a larger number of patients.
- Lines 63- 65: You state that on 2019 evaluation, the patient remained on WHO-FC II-late status and RHC revealed the following deterioration: mPAP was increased to 55 mmHg; PVR was increased to 9.2 Wood Units while CI was decreased to 2.7 lt/ min/m2.
Answer: You are totally right. Text was rephrased according the correct meaning of the RHC results compared to 2019.
- Lines 70-73: Here you state that during the last evaluation on October 2020, we noticed that the patient’s WHO-FC status was upgraded to II-early. RHC confirmed that her hemodynamic parameters were improved: mPAP was decreased to 48 mmHg; PVR was decreased to 6.9 Wood Units and CI was increased to 2.7 lt/ min/m2. The figure shows the CI to be 3L/min/m2. Please correct this discrepancy.
Answer: Once again we have to thank you for looking at the Figure and understanding that text was misleading and not properly referred to correct RHC numbers. We are very sorry for this and we thank you for your review comments.
Round 2
Reviewer 1 Report
I would like to thank the authors for addressing my concerns. The presented data is definitely of interest. I have only a minor suggestion to make.
In my opinion, all PBR values (PBR5-9, PBR 10-19, PBR 20-25, PBR 5-25), and maybe also RBC filling should be reported, even if the values do not show any difference. Reporting only PBR 5-9 can lead to a certain bias, if a capillary loss is present in this population. If that's the case, the damaged capillaries are not perfused (where the glycocalyx is probably also damaged) and cannot be measured with GlycoCheck. Therefore, reporting the whole spectrum of 5-25um, as well as RBC filling might be of value.
Author Response
Reviewer's comments
I would like to thank the authors for addressing my concerns. The presented data is definitely of interest. I have only a minor suggestion to make. In my opinion, all PBR values (PBR5-9, PBR 10-19, PBR 20-25, PBR 5-25), and maybe also RBC filling should be reported, even if the values do not show any difference. Reporting only PBR 5-9 can lead to a certain bias, if a capillary loss is present in this population. If that's the case, the damaged capillaries are not perfused (where the glycocalyx is probably also damaged) and cannot be measured with GlycoCheck. Therefore, reporting the whole spectrum of 5-25um, as well as RBC filling might be of value.
Answer: Thank you for giving us the opportunity to add more data in order to strengthen our results (all the data you asked for are already added in the most update version of our article).